# Targeted Active Learning for Bayesian Decision-Making

**Louis Filstroff**                    *louis.filstroff@centralelille.fr*
*Univ. Lille, CNRS, Centrale Lille, UMR 9189 CRIStAL, F-59000 Lille, France*

**Iiris Sundin**                    *iiris.sundin@iki.fi*
*Department of Computer Science*
*Aalto University, Finland*

**Petrus Mikkola**                    *petrus.mikkola@gmail.com*
*Department of Computer Science*
*Aalto University, Finland*
*University of Helsinki, Finland*

**Aleksei Tiulpin**
*Research Unit of Health Sciences and Technology*
*University of Oulu, Finland*

**Juuso Kylmäoja**                    *juuso.kylma@gmail.com*
*Department of Computer Science*
*Aalto University, Finland*

**Samuel Kaski**                    *samuel.kaski@aalto.fi*
*Department of Computer Science*
*Aalto University, Finland*

**Reviewed on OpenReview:** *https://openreview.net/forum?id=KxPjuiMgmm*

## Abstract

Active learning is usually applied to acquire labels of informative data points in supervised learning, to maximize accuracy in a sample-efficient way. However, maximizing the supervised learning accuracy is not the end goal when the results are used for decision-making, for example in personalized medicine or economics. We argue that when acquiring samples sequentially, the common practice of separating learning and decision-making is sub-optimal, and we introduce an active learning strategy that takes the down-the-line decision problem into account. Specifically, we adopt a Bayesian experimental design approach, in which the proposed acquisition criterion maximizes the expected information gain on the posterior distribution of the optimal decision. We compare our targeted active learning strategy to existing alternatives on both simulated and real data and show improved performance in decision-making accuracy.

## 1 Introduction

Supervised learning techniques aim at learning a function that maps the input $\mathbf{x} \in \mathcal{X}$ to the outcome (or label) $y \in \mathcal{Y}$, based on a collection of examples $\mathcal{D} = \{(\mathbf{x}_i, y_i)\}_{i=1}^N$. Whereas having access to thousands of unlabeled data is nowadays easy, obtaining the associated labels is expensive in many applications, such as those involving human experts (e.g., image annotating), or running additional experiments. In this context, *active learning* (AL) aims at iteratively querying for the most informative data point among a pool of unlabeled data (Settles, 2012). In the machine learning literature, the term "active learning" often implies a classification task, but the concept straightforwardly extends to regression, and the same problem arises in

the statistics literature under the names "optimal experimental design" or "Bayesian experimental design" (BED) (Chaloner & Verdinelli, 1995; Ryan et al., 2016).

Active learning boils down to the selection criterion for the next point to label. Popular strategies include uncertainty sampling (Lewis & Catlett, 1994), expected error reduction (Roy & McCallum, 2001), or expected information gain on the model parameters (Lindley, 1956; MacKay, 1992). All these strategies aim at learning a model as accurately as possible with as few queries as possible. However, the accuracy of the model is not the end goal in all scenarios. In this paper, we consider the setting where the model is used for decision-making. Each action (or decision) is assessed by its utility, and the optimal decision is the one that yields the highest expected utility.

We focus on a decision-making task where the goal is to select the optimal decision for a single test point, while we also extend our method to the case of a test population. Such a scenario arises, for instance, in the topical field of personalized medicine. Based on the history of previous patients, described by patient covariates, the treatments they received, and the observed outcomes, a model is built to infer the individualized treatment effect (Wager & Athey, 2018; Shalit et al., 2017; Alaa & van der Schaar, 2017; Yao et al., 2018; Bica et al., 2020). A doctor will then use the model's predictions to choose the best treatment for a new patient. In this motivational example, the active learning task addressed in our paper involves identifying the most informative patient-treatment pairs from a predefined pool, which may include sources such as electronic health records (EHR). Accessing data from these records involves stringent legal justification due to privacy concerns, mandating that a physician must have a legitimate reason for retrieval and use in patient treatment. The access process can be lengthy and incur significant financial costs, or the administrative costs per patient can quickly accumulate, becoming intolerable for large datasets.

Traditionally, model learning and decision-making are carried out separately, i.e., the learning phase is blind to the decision-making problem. This is sub-optimal when data can be collected actively, and as such, there is a need for active learning strategies that take into account this downstream decision-making task. This problem of *decision-making-aware* active learning has recently received attention by Sundin et al. (2019), who proposed a heuristic strategy for a binary decision-making problem. However, their criterion does not extend to more complex situations, such as multiple-decision problems, which limits applications.

In this paper, we propose a principled selection criterion for decision-making-aware active learning. More precisely, we adopt a BED approach, where it is a well-known fact that the optimal strategy is to perform expected information gain on the quantity of interest (Chaloner & Verdinelli, 1995). In our setting, we identify that quantity as the *epistemic uncertainty on the optimal decision*; in other words, the proposed criterion will aim at maximally reducing the uncertainty of the posterior distribution of the optimal decision. This is unlike classical BED approaches, which either target model parameters or outcomes. The effect of the proposed methodology versus classical ones is illustrated in Figure 1. Our active learning criterion is *optimal* in the sense that it maximally reduces uncertainty about which decision yields the highest expected utility for a given test point.

We consider *performance improvement at first $t$* additional acquisitions for small $t$, which is a key measure for scenarios where acquisitions are costly, such as personalized medicine and generally having a human in the loop. Specifically, our method is applicable when a) the test population of the decision-making task is known: a single individual $\tilde{\mathbf{x}}$ (as in personalized medicine), or a set of individuals, and b) it is possible to collect more data on-demand, for example, once the model has been deployed, but c) these queries are costly due to, e.g., requesting new experiments, involving experts, or fulfilling privacy constraints. We empirically demonstrate the advantages of the proposed method with respect to existing AL baselines, both in simulated and real-world experiments.

## 2 Problem formulation

### 2.1 Modeling of outcomes

We consider a regression setting with covariates $\mathbf{x} \in \mathbb{R}^p$ and outcomes $y \in \mathbb{R}$. We further assume that the outcome also depends on a decision variable $d \in \{1, \dots K\}$. Typically, the outcome is observed after an action

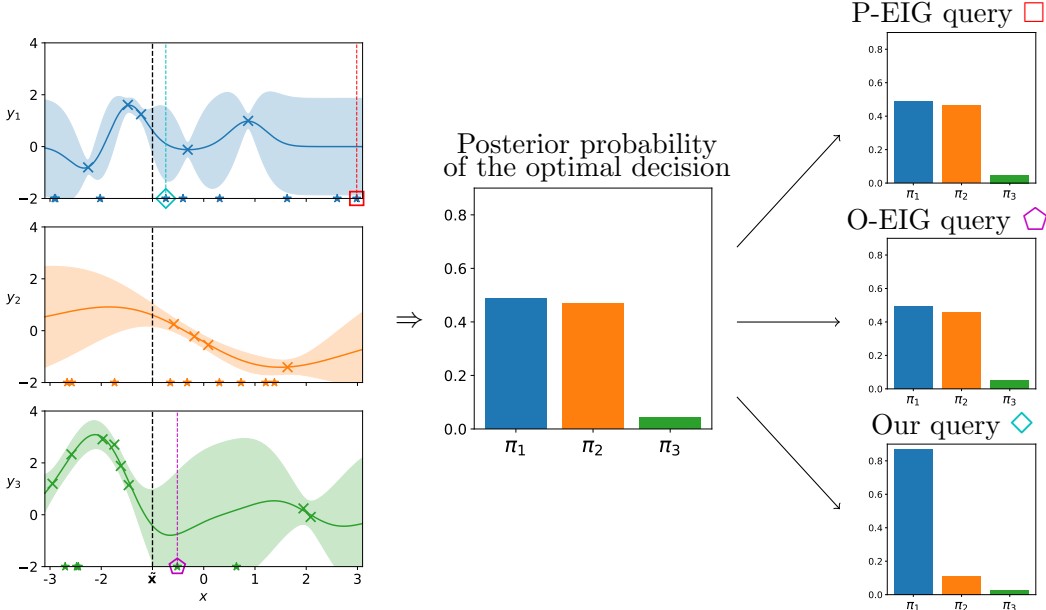

Figure 1: Illustrative example of a decision-making task: choose one decision from $d = \{1, 2, 3\}$ at point $\tilde{\mathbf{x}} = -1$ (black dashed line). (Left): The function $f_k$ models the outcome for a decision $d = k$ based on a learning dataset (colored lines with uncertainty intervals). The learning dataset consists of labeled ($\times$ marker) and unlabeled ($\star$ marker) data points. (Center): The posterior distribution of the optimal decision helps in making the decision (here the Bayes-optimal decision is $d = 1$), and assessing its uncertainty. (Right): Evolution of that distribution after querying one additional point. Using the standard EIG criteria (Sec. 2.3) does not help the decision-making (top, middle), while the proposed targeted AL criterion greatly improves it by reducing its uncertainty (bottom).

has been taken. In the healthcare application, this corresponds to observing the effect of treatment $d$ on a patient. We therefore have a training set $\mathcal{D}$ comprising triplets, i.e., $\mathcal{D} = \{(x_i, d_i, y_i)\}_{i=1}^N$.

Denoting by $y_k$ the variable $y|(d = k)$, the goal is therefore to learn the functions $f_k$ which map $\mathbf{x}$ to $y_k$. In this work, we assume that the $y_k$ are conditionally independent given $\mathbf{x}$, and write

$$y_k = f_k(\mathbf{x}) + \epsilon_k, \tag{1}$$

where $\epsilon_k \sim \mathcal{N}(0, \sigma_k^2)$.

Moreover, we assume that we are equipped with a functional prior distribution on $f_k$, such as a Gaussian process (GP) or a Bayesian neural network, which in turn allows us to deal with posterior uncertainty. Indeed, given $\mathcal{D}_k = \{(x_i, d_i, y_i) \in \mathcal{D} \mid d_i = k\}$, and using the notation $f_{k,\mathbf{x}}$ to denote $f_k(\mathbf{x})$, we may characterize the posterior distribution $p(f_{k,\mathbf{x}}|\mathcal{D}_k)$ for all $\mathbf{x}$. Lastly, we treat $\sigma_k^2$ as a hyperparameter, to be estimated with, e.g., maximum marginal likelihood.

## 2.2 Decision-making problem

For clarity of presentation, when introducing the method we will focus on a single test input, which we denote by $\tilde{\mathbf{x}}$, rather than a test population. Nevertheless, the developments presented in the paper can straightforwardly be extended to a test population, as we will describe in Section 3.2.

The input $\tilde{\mathbf{x}}$ is a previously unseen data point for which the end-user of the model has to make a decision, i.e., has to choose one among the set of the $K$ available decisions. In our introductory example, $\tilde{\mathbf{x}}$ corresponds to the covariates of a patient for whom the doctor has to choose a treatment.

In this context, decisions are assessed through a scalar utility: the higher the better. Utilities can be computed from the outcomes $\tilde{y}_k$; we write $u_k = r_k(\tilde{y}_k)$, where the $r_k$ are known, deterministic functions that map the outcomes to the utilities. In the remainder of the paper, we will assume, without loss of generality, that $u = y$.

Given that the models have been trained on $\mathcal{D} = \cup_k \mathcal{D}_k$, we assume that the user behaves optimally in the sense of (evidential) decision theory, i.e., chooses the decision which yields the greatest expected utility at $\tilde{\mathbf{x}}$. The Bayes-optimal decision[1] is

$$d_{\mathrm{BAYES}} = \underset{k \in \{1,\ldots,K\}}{\arg\max} \iint \tilde{y}_k p(\tilde{y}_k|f_{k,\tilde{\mathbf{x}}}) p(f_{k,\tilde{\mathbf{x}}}|\mathcal{D}_k) \mathrm{d}f_{k,\tilde{\mathbf{x}}} \mathrm{d}\tilde{y}_k. \tag{2}$$

## 2.3 Bayesian active learning

We assume access to a pool of unlabeled data $\mathcal{U} = \{(\mathbf{x}_j, d_j)\}_{j=1}^J$, from which the associated outcomes can be actively queried. We wish to select queries from $\mathcal{U}$ which are maximally useful for the decision-making problem, i.e., queries that reduce uncertainty on the optimal decision for $\tilde{\mathbf{x}}$. Our primary example involves a scenario where information about the outcome for an unlabeled data point can be retrieved from a database (e.g. containing Electronic Health Records, EHR).

**Conventional Bayesian experimental design strategies**

Let us consider a standard Bayesian regression formulation, i.e., when the relationship between the input $\mathbf{x}$ and the outcome $y$ is modeled by a likelihood $p(y|\mathbf{x}, \theta)$, where $\theta$ are latent parameters with a prior distribution $p(\theta)$. We wish to decide on the next point to query. The principled strategy from an information-theoretic perspective is to look for the query that maximizes the expected information gain (EIG) on a quantity of interest, which is defined as the expected reduction of the entropy of the posterior distribution of the quantity of interest.

---

[1]Due to the assumption of an identity utility function $u = y$, this corresponds to risk-neutral Bayes-optimal decision.

Typically, the parameters $\theta$ are chosen to be that quantity; we refer to that strategy as `P-EIG`. In that case, the optimal query is such that

$$\mathbf{x}_{\text{P-EIG}}^{\star} = \underset{\mathbf{x}_j \in \mathcal{X}}{\arg\max} \left( \mathcal{H}[p(\theta|\mathcal{D})] - \mathbb{E}_{p(y_j|\mathbf{x}_j,\mathcal{D})} \big[ \mathcal{H}[p(\theta|\mathcal{D} \cup \{(\mathbf{x}_j, y_j)\})] \big] \right), \tag{3}$$

where we used the notation $\mathcal{H}[p(.)]$ to denote the differential entropy of a probability distribution (Shannon, 1948). This idea was first suggested by Lindley (1956), and has then been considered by several authors, see for example Bernardo (1979); MacKay (1992); Houlsby et al. (2011); Hernández-Lobato et al. (2014). Moreover, the criterion of Eq. (3) can be rearranged in a form that computes entropies in the outcome space rather than the parameter space (this has been coined "BALD" by Houlsby et al. (2011)), which allows us to define it in a non-parametric setting (i.e., when $y = f(\mathbf{x}) + \epsilon$):

$$\mathbf{x}_{\text{P-EIG}}^{\star} = \underset{\mathbf{x}_j \in \mathcal{X}}{\arg\max} \left( \mathcal{H}[p(y_j|\mathbf{x}_j,\mathcal{D})] - \mathbb{E}_{p(f|\mathcal{D})} \big[ \mathcal{H}[p(y_j|\mathbf{x}_j,f)] \big] \right), \tag{4}$$

see details on the equivalence in Appendix B. Nonetheless, the EIG remains a challenging criterion to compute, as it involves nested Monte Carlo estimation (Rainforth et al., 2018), and several recent works have aimed at mitigating this issue (Foster et al., 2019; Zheng et al., 2020).

The other standard option is to consider the quantity of interest to be $\tilde{y}$, the outcome at a specific $\tilde{\mathbf{x}}$ (i.e., not belonging to the unlabeled set) (Krause et al., 2008; Daee et al., 2017; Sundin et al., 2018). Let us refer to that strategy as `O-EIG`. The optimal query becomes

$$\mathbf{x}_{\text{O-EIG}}^{\star} = \underset{\mathbf{x}_j \in \mathcal{X}}{\arg\max} \left( \mathcal{H}[p(\tilde{y}|\tilde{\mathbf{x}},\mathcal{D})] - \mathbb{E}_{p(y_j|\mathbf{x}_j,\mathcal{D})} \big[ \mathcal{H}[p(\tilde{y}|\tilde{\mathbf{x}},\mathcal{D} \cup \{(\mathbf{x}_j, y_j)\})] \big] \right). \tag{5}$$

**Shortcomings for the decision-making problem**

The conventional Bayesian AL criteria of Eqs. (4) and (5) can easily be adapted to our setting, and would select the element of $\mathcal{U}$ which yields the highest information gain over any of the $f_k$, or over any of the $\tilde{y}_k$, respectively. However, such queries are not necessarily helpful to improve the quality of the decision-making. Indeed, they may have little to no impact on the posterior predictive distributions at $\tilde{\mathbf{x}}$, or, may improve predictions only for a decision that has very little probability of being the optimal one. Such phenomena are displayed on the right panel of Figure 1.

Thus, we present in the next section an active learning strategy which takes the decision-making problem into account by considering the posterior distribution of the optimal decision for $\tilde{\mathbf{x}}$, and which therefore overcomes the aforementioned shortcomings.

## 3 Targeted active learning criterion

### 3.1 Posterior uncertainty on the optimal decision

The optimal decision is, by definition, the one with the highest expected utility. If we knew the value of $f_{k,\tilde{\mathbf{x}}}$ exactly for all $k$, then the optimal decision would be known with 100% certainty. However, since we work with a finite sample size, we cannot have access to the value of $f_{k,\tilde{\mathbf{x}}}$, but instead, we characterize posterior distributions $p(f_{k,\tilde{\mathbf{x}}}|\mathcal{D}_k)$, which in turn leads to the Bayes-optimal recommendation of Eq. (2). As it turns out, by fully taking advantage of the Bayesian framework, we can go beyond mere point recommendation and estimate the posterior uncertainty that decision $k$ is the optimal decision (at $\tilde{\mathbf{x}}$).

Let us denote by $\pi_k$ the posterior probability that decision $k$ is the optimal decision. We further define $D_{\text{best}}(\tilde{\mathbf{x}})$ to be the discrete random variable whose probability mass function is given by the $(\pi_k)_{k=1}^K$. In other words, $D_{\text{best}}(\tilde{\mathbf{x}})$ contains the posterior uncertainty on the optimal decision for $\tilde{\mathbf{x}}$. We have

$$\pi_k = \mathbb{P}\left( \mathbb{E}(\tilde{y}_k|f_{k,\tilde{\mathbf{x}}}) = \max_{k'} \mathbb{E}(\tilde{y}_{k'}|f_{k',\tilde{\mathbf{x}}}) \right). \tag{6}$$

In the equation above, the conditional expectations are to be understood as random variables, and $f_{k,\tilde{\mathbf{x}}} \sim p(f_{k,\tilde{\mathbf{x}}}|\mathcal{D}_k)$. Given the model assumptions, it turns out that $\mathbb{E}(\tilde{y}_k|f_{k,\tilde{\mathbf{x}}}) = f_{k,\tilde{\mathbf{x}}}$. We can further write

$$\pi_k = \mathbb{P}\left(f_{k,\tilde{\mathbf{x}}} = \max_{k'} f_{k',\tilde{\mathbf{x}}}\right), \tag{7}$$

$$= \mathbb{P}\left(\bigcap_{k' \neq k} \{f_{k,\tilde{\mathbf{x}}} > f_{k',\tilde{\mathbf{x}}}\}\right). \tag{8}$$

The events inside Eq. (8) are not independent, and as such this cannot be broken down into a product of probabilities. An illustrative problem with 3 decisions is displayed on Figure 1, with the current models in the left panel, and the associated probabilities $\pi_k$ in the middle panel.

It is important to note that the randomness of $f_{k,\tilde{\mathbf{x}}}$ only comes from lack of information. Such uncertainty is said to be *epistemic*, and adding more points to the dataset will reduce this uncertainty (Hüllermeier & Waegeman, 2021). Therefore, it is more precise to characterize $D_{\text{best}}(\tilde{\mathbf{x}})$ as the random variable containing the epistemic uncertainty on the optimal decision. We argue that this is the variable of interest in our setting.

### 3.2 Decision-targeted active learning criterion

Now that we have characterized the posterior distribution of interest (the distribution of the variable we called $D_{\text{best}}(\tilde{\mathbf{x}})$), we propose to sequentially select the data point from $\mathcal{U}$ which maximizes the expected information gain about this posterior distribution. We write

$$(\mathbf{x}^\star, d^\star) = \underset{(\mathbf{x}_j, d_j) \in \mathcal{U}}{\arg\max} \Bigg( \mathcal{H}[p(D_{\text{best}}(\tilde{\mathbf{x}})|\mathcal{D})] \tag{9}$$

$$- \mathbb{E}_{p(y_{d_j}|\mathbf{x}_j, \mathcal{D}_{d_j})} \left[\mathcal{H}[p(D_{\text{best}}(\tilde{\mathbf{x}})|\mathcal{D} \cup \{(\mathbf{x}_j, d_j, y_{d_j})\})]\right] \Bigg),$$

which means that these queries aim at reducing the uncertainty on the optimal decision of $\tilde{\mathbf{x}}$. The criterion of Eq. (9) may be rewritten in a simpler form as

$$(\mathbf{x}^\star, d^\star) = \underset{(\mathbf{x}_j, d_j) \in \mathcal{U}}{\arg\min} \mathbb{E}_{p(y_{d_j}|\mathbf{x}_j, \mathcal{D}_{d_j})} \left[\mathcal{H}[p(D_{\text{best}}(\tilde{\mathbf{x}})|\mathcal{D} \cup \{(\mathbf{x}_j, d_j, y_{d_j})\})]\right]. \tag{10}$$

The full decision-making-aware active learning process is illustrated on Figure 2.

**Extension to a test population**

We briefly consider here the scenario where there is a collection of previously unseen testing points $(\tilde{\mathbf{x}}_i)_{i=1}^{N_t}$, for which we wish to improve the decision-making. The optimal decision for $\tilde{\mathbf{x}}_i$ is to be understood, as before, as the one which yields the highest expected utility at $\tilde{\mathbf{x}}_i$. Similarly, we can define $D_{\text{best}}(\tilde{\mathbf{x}}_i)$ as the discrete random variable which contains the (epistemic) uncertainty on the optimal decision for each $\mathbf{x}_i$.

The extension of Eq. (10) simply consists in considering the entropy of the joint posterior distribution of $D_{\text{best}}(\tilde{\mathbf{x}}_1), \ldots, D_{\text{best}}(\tilde{\mathbf{x}}_{N_t})$. However, this quickly leads to computational issues, as the cardinality of the space is now $K^{N_t}$. To alleviate this issue, we propose to minimize an upper bound of the entropy instead by applying the chain rule for differential entropy, which leads to the following criterion

$$(\mathbf{x}^\star, d^\star) = \underset{(\mathbf{x}_j, d_j) \in \mathcal{U}}{\arg\min} \mathbb{E}_{p(y_{d_j}|\mathbf{x}_j, \mathcal{D}_{d_j})} \left[\sum_{i=1}^{N_t} \mathcal{H}[p(D_{\text{best}}(\tilde{\mathbf{x}}_i)|\mathcal{D} \cup \{(\mathbf{x}_j, d_j, y_{d_j})\})]\right]. \tag{11}$$

The tightness of the bound depends on the degree of dependence between the optimal decisions and the test points. For instance, if the optimal treatment is similar for all patients, the bound is not very tight. Conversely, if the optimal treatments are highly specific to individual patients, the bound becomes significantly tighter.

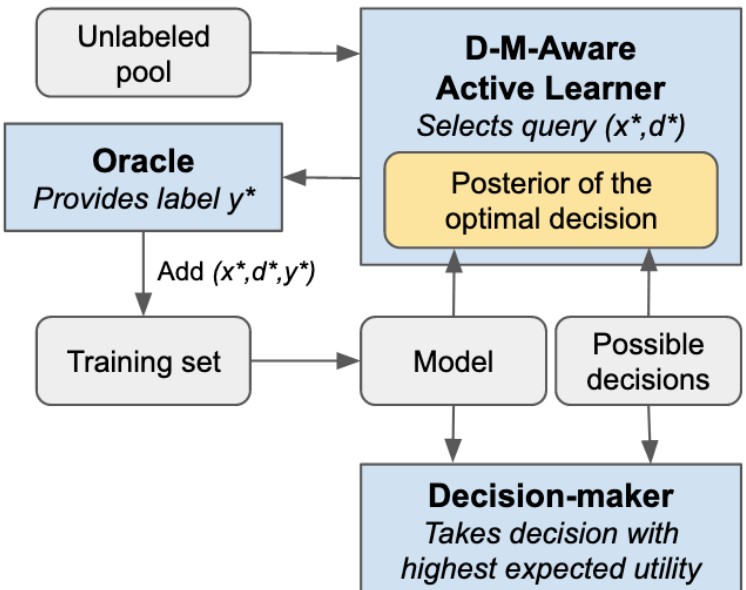

Figure 2: Decision-making-aware active learning. Agents are in blue boxes. The active learner is aware of the down-the-line decision-making problem, and selects targeted queries for the problem, by taking into account the posterior distribution of the optimal decision. Once the learning phase is over, we here assume that the end-user takes the action that yields the highest expected utility.

### 3.3 Practical implementation

Computing the criterion Eq. (10) requires to solve two computational challenges:

1. The expectation w.r.t. $p(y_{d_j}|\mathbf{x}_j, \mathcal{D}_{d_j})$ is intractable and needs to be approximated;

2. The probabilities $\pi_d$ are not known in closed form either. They need to be estimated in order to compute the entropy of $D_{\text{best}}(\tilde{\mathbf{x}})$.

To approximate the expectation, we resort to Monte Carlo approximation. This means that given $N_s$ samples $y_{d_j}^{(l)}$ drawn from $p(y_{d_j}|\mathbf{x}_j, \mathcal{D}_{d_j})$, we have

$$\mathbb{E}_{p(y_{d_j}|\mathbf{x}_j, \mathcal{D}_{d_j})} \left[ \mathcal{H}[p(D_{\text{best}}(\tilde{\mathbf{x}})|\mathcal{D} \cup \{(\mathbf{x}_j, d_j, y_{d_j})\})] \right]$$

$$\simeq \frac{1}{N_s} \sum_{l=1}^{N_s} \mathcal{H}[p(D_{\text{best}}(\tilde{\mathbf{x}})|\mathcal{D} \cup \{(\mathbf{x}_j, d_j, y_{d_j}^{(l)})\})]. \tag{12}$$

Note that when $p(y_{d_j}|\mathbf{x}_j, \mathcal{D}_{d_j})$ is Gaussian, as is the case in GP regression, we may use a Gauss-Hermite approximation scheme (see Appendix C).

Next, to compute the entropy $\mathcal{H}[p(D_{\text{best}}(\tilde{\mathbf{x}})|\mathcal{D})]$, we need to know the posterior probabilities $\pi_k$ (given by Eq. (8)). Unfortunately, closed-form solutions do not exist in general, and we resort to a straightforward approximation scheme. We take sets of posterior draws from $p(f_{k,\tilde{\mathbf{x}}}|\mathcal{D}_k)$ for all $k$ to generate posterior samples of $D_{\text{best}}(\tilde{\mathbf{x}})$, which are then used to estimate the entropy. For simplicity, we estimate the entropy of the multinomial distribution by using empirical estimates of the $\pi_k$ from the posterior samples of $D_{\text{best}}(\tilde{\mathbf{x}})$.

Pseudo-code of the algorithm computing the proposed targeted AL criterion is given in Algorithm 1. All the computational burden resides in the model retraining step, which has to be carried out $N_s \times \text{card}(\mathcal{U})$ times to solve the optimization problem in Eq. (10). The computational complexity is high, but many operations are trivially parallelizable, for example over all elements of $\mathcal{U}$, or even over all Monte Carlo samples. Moreover,

pre-selection strategies may be implemented to avoid computing the criterion for all elements of $\mathcal{U}$, or the selection problem itself could be cast as a Bayesian optimization problem.

---

**Algorithm 1** Estimating the criterion Eq. (10) for $(\mathbf{x}_j, d_j) \in \mathcal{U}$

---

   $C = 0;$                                                                       $\triangleright$ *Current estimate of Eq.* (10)
   $\triangleright$ *Monte Carlo approximation*
   **for** $l = 1, \dots, N_s$ **do**
      $y_{d_j}^{(l)} \sim p(y_{d_j} | \mathbf{x}_j, \mathcal{D}_{d_j})$
      Add $(\mathbf{x}_j, y_{d_j}^{(l)})$ to the training set $\mathcal{D}_{d_j}$
      Optimize GP hyperparemeters associated with decision $d_j$
      $\triangleright$ *Estimate entropy of* $D_{best}(\tilde{\mathbf{x}})$ *with augmented dataset*
      **for** $k = 1, \dots, K$ **do**
         Get samples from $p(f_{k,\tilde{\mathbf{x}}} | \mathcal{D}_k)$
      **end for**
      Compute estimates of the $\pi_k$
      Compute entropy $\mathcal{H}$ from the $\pi_k$
      $C = C + N_s^{-1}\mathcal{H}$
      Remove $(\mathbf{x}_j, y_{k,j}^{(l)})$ from the training set $\mathcal{D}_{d_j}$
   **end for**

---

Lastly, we emphasize that working with a test population, i.e., dealing with Eq. (11) instead of Eq. (10), brings negligible additional computational complexity. Indeed, the only difference is that we would have to estimate several entropy values instead of one, which has a negligible cost compared to retraining the model, as previously stated.

## 4 Related work

**Decision-making-aware strategies in machine learning.** We begin by discussing such strategies in a *passive* learning context. Lacoste-Julien et al. (2011) introduced the so-called loss-calibrated inference framework. They characterized the decision-making problem by a loss (i.e., negative utility), which is taken into account to alter the learning objective of variational inference. This work has been extended, e.g., to Bayesian neural networks (Cobb et al., 2018) and to continuous decisions (Kuśmierczyk et al., 2019). Another line of work, which tackles the computation of expected functions (w.r.t. a posterior distribution), is discussed by Rainforth et al. (2020). The authors argued that when these functions are known in advance, it is beneficial to take them into account and subsequently proposed a framework coined TABI (target-aware Bayesian inference), which enables efficient estimation of such quantities.

Surprisingly enough, the literature is quite sparse when it comes to similar strategies for active learning. Saar-Tsechansky & Provost (2007) proposed two heuristics to help choosing which customers to target in marketing campaigns. More recently, Sundin et al. (2019) proposed an active learning criterion based on the Type-S error to improve binary decisions. Several recent works tackled goal-oriented active learning, but none of those consider the decision-making step that comes after the learning process. For instance, Yan et al. (2018) proposed a debiasing query strategy based on disagreement-based active learning when learning classifiers from logged data (where the labels have been revealed according to a logging policy, leading to biased training sets). Their work is also limited to binary decisions. Kandasamy et al. (2019) introduced a reward function and a method based on posterior sampling, and Xu & Kazantsev (2019) introduced a utility function and the use of so-called influence functions, but the words "reward" or "utility" there refer to different metrics of model evaluation. Finally, Zhao et al. (2021) proposed an uncertainty-aware AL criterion for classification with 0-1 loss, which focuses only on the reduction of the uncertainty that pertains to the classification error.

**Best arm identification in multi-armed bandits.** The decision-making problem we consider can equivalently be presented as the problem of identifying which of the $K$ arms, described by the distributions

of the utilities of each decision at $\tilde{\mathbf{x}}$, is the best (i.e., yields the highest expected utility, or reward). This is known in the multi-armed bandits literature as the "best arm identification" problem, or "pure exploration" problem, which has been studied both from frequentist and Bayesian perspectives (Audibert et al., 2010; Kaufmann et al., 2016; Russo, 2016). A generalization of this problem has recently been introduced under the name "transductive bandits" (Fiez et al., 2019). The objective of such problems differs from the traditional setting of multi-armed bandits, which is to maximize the cumulative sum of rewards.

However, the setting of best arm identification problems differs from ours in the possible arms that can be sampled. In contrast to these problems, we cannot sample from the different arms at $\tilde{\mathbf{x}}$. We can only sample once from a specific set arms defined by the pairs $(\mathbf{x}, d) \in \mathcal{U}$. This prevents us from using strategies from the multi-armed bandits literature. Instead, by adding new points to the regression models, we aim at better characterizing the distributions of the expected utilities at $\tilde{\mathbf{x}}$.

**Bayesian optimization and active learning.** Bayesian optimization (BO) refers to a class of algorithms for global optimization of black-box functions, where a probabilistic surrogate model such as a Gaussian process is placed on the objective function (Brochu et al., 2010; Garnett, 2023). BO algorithms sequentially select points where the objective function is evaluated, based on some acquisition function that typically balances exploration and exploitation. As such, BO is closely related to AL, see for example Ling et al. (2016) for a unifying framework of some standard AL and BO algorithms. Conceptually, BO can be seen as a goal-oriented AL strategy, but for the specific decision-making problem of choosing a point that maximizes a black-box function. Many acquisition functions can be derived using Bayesian decision theory by choosing an appropriate definition for the utility of the gathered dataset $u(\mathcal{D})$ (Garnett, 2023), which should not be confused with the utility of the outcome discussed in Section 2.2.

**Entropy-search multi-fidelity Bayesian optimization** Multi-fidelity Bayesian Optimization (MFBO) extends traditional BO by incorporating auxiliary information sources, also termed as low-fidelities (Kandasamy et al., 2016). These fidelities offer a cost-effective means of gathering insights about the objective function, which serves as the primary information source. A strong link exists between the proposed method and an entropy-search based MFBO, which we will explore next.

Consider a family of functions $f_\mathbf{x} : \llbracket K \rrbracket \to \mathbb{R}$, indexed by the information sources $\mathbf{x} \in \mathcal{X}$. The primary information source and the corresponding objective function can be defined as $\mathbf{x}_{test}$ and $f_{\mathbf{x}_{test}}$, respectively. The auxiliary information sources and their associated functions are denoted by $\mathbf{x} \neq \mathbf{x}_{test}$ and $f_{\mathbf{x} \neq \mathbf{x}_{test}}$, respectively. Our restriction that querying $f_{\mathbf{x}_{test}}(d)$ for any $d \in \llbracket K \rrbracket$ is not possible can be interpreted as the cost of $f_{\mathbf{x}_{test}}(d)$ always exceeding the total budget. This restriction also applies to any query $(\mathbf{x}, d) \notin \mathcal{U}$. For any $(\mathbf{x}, d) \in \mathcal{U}$, the cost is a constant $budget/T$ where $T$ is the total number of iterations. The MFBO acquisition problem can thus be formulated as follows: Which information source $\mathbf{x}$ and which point $d$ should be chosen in order to maximally assist in finding $\arg\max_{d \in \{1,...,K\}} f_{\mathbf{x}_{test}}(d)$?

If we adopt a policy aiming to maximize the information gain of $\arg\max_{d \in \{1,...,K\}} f_{\mathbf{x}_{test}}(d)$ (i.e., we set $\omega = \arg\max_{d \in \{1,...,K\}} f_{\mathbf{x}_{test}}(d)$ as Garnett (2023), Equation (6.8)), we are close to the Predictive Entropy Search (PES) method proposed by Hernández-Lobato et al. (2014). However, PES was initially formulated in a single-fidelity BO setting. In the literature, there are extensions of entropy-search based methods to multi-fidelity settings, such as the Multi-Fidelity Maximum-Entropy-Search (MF-MES) method introduced by Takeno et al. (2020). Therefore, our acquisition criterion (9) can be interpreted as an MF-PES method. While this specific method is not directly identified in existing literature, it can be conceptualized as a combination of PES (Hernández-Lobato et al., 2014, Equation (2)), and for instance MF-MES (Takeno et al., 2020, Supplement, Equation (18)).

What remains to be addressed is the design of the multi-fidelity GP. The MFGP kernel can be specified as $k((\mathbf{x}, d), (\mathbf{x}', d')) = \sum_d \kappa_d(\mathbf{x}, \mathbf{x}')\kappa(d, d')$, where $\kappa(d, d') = \mathbb{I}(d = d')$ and $\kappa_d(\mathbf{x}, \mathbf{x}')$ corresponds to the kernel chosen in Section 5.2. With this MFGP model, our method aligns with MF-PES when the multi-fidelity setting is framed as described above. It is noteworthy that the derived MFGP kernel is unconventional, as it treats the information source kernel $\kappa_d(\mathbf{x}, \mathbf{x}')$ as dependent on the data point, with distinct hyperparameters for each $d \in \{1,...,K\}$. This flexibility allows for more nuanced learning of the correlations between various

information sources $\mathbf{x}$ and each $d$. We hypothesize that this inductive bias, combined with a decision-focused ($d$-focused) acquisition criterion, may contribute to the superior performance observed in our experiments.

## 5 Experiments

### 5.1 Use-cases and datasets

**Fully synthetic data.** We proceed to generate a dataset of 400 points of dimension 5. The covariates are drawn from the standardized Gaussian distribution. We generate four different outcomes as independent realizations of GPs with squared exponential kernels whose variance and lengthscales are different. These outcomes are then corrupted by Gaussian white noise. Finally, the decision variable associated with each point is drawn randomly, but not uniformly, to mimic the imbalance in treatment assignment.

**Treatment recommendation.** The first use-case focuses on the topical personalized medicine research question of using electronic health records (EHRs) to augment data from randomized controlled trials (RCT). In this setting, the training set contains individuals $\mathbf{x}$, and the outcome $y$ of the treatment $d$ that they received. In addition, we assume a record of patients and treatments, for which the outcomes can be acquired. An example case is EHRs that contain information about the prescription of treatment without follow-up, in which case a new appointment or call needs to be scheduled with the patient in order to acquire the outcome. The objective is to improve the decision of which treatment to give to a new patient $\tilde{\mathbf{x}}$, as in the study of Sundin et al. (2019).

Experiments are run on the IHDP dataset[2] (Hill, 2011), a semi-synthetic dataset which consists of 747 patients with 25 covariates. The patient covariates come from a real randomized medical study from the 80s, however, the outcomes have been artificially generated, implying that all potential outcomes are available. We combine the responses A1, B1, and C1 to obtain a 3-decision problem.

**Knee osteoarthritis diagnosis.** The second use case focuses on symptomatic patients who have a suspicion of knee osteoarthritis (OA) progression in the medial compartment of the right knee. OA is a degenerative disorder of the joints, which reveals itself through symptomatic and structural changes. To date, this disease has no cure, but if detected early, its progression could be slowed down via behavioral interventions (Katz et al., 2021). We thus consider the problem of optimizing the diagnostic path for a new patient $\tilde{\mathbf{x}}$. More precisely, the decision-making problem is to decide when to perform the next follow-up: at 12, 24, 36 or 48 months, or after 48 months. We assume that the doctor is able to query for additional data about previous patients, but that requires a laborious authorization process due to privacy concerns.

We construct a dataset from the Osteoarthritis Initiative (OAI) database[3], which is a multi-center 10-year observational longitudinal study of 4796 subjects (consent obtained from all the subjects; data are de-identified). After pre-processing, we obtain our final dataset with 8 covariates (clinical data and an initial imaging-based assessment) from 606 patients. The outcome is the joint space width loss over 0.7mm by the time of the follow-up (Neumann et al., 2009; Eckstein et al., 2015). A detailed description of the dataset is given in Appendix D.

### 5.2 Model of the outcomes

All experiments are run with GP regression (Rasmussen & Williams, 2006), i.e., we assume a zero-mean GP prior for the function $f_k$, with kernel $\kappa_k$:

$$f_k \sim \mathrm{GP}(\mathbf{0}, \kappa_k(\mathbf{x}, \mathbf{x}')). \tag{13}$$

Note that in this case, posterior distributions $p(f_{k,\tilde{\mathbf{x}}}|\mathcal{D}_k)$ turn out to be Gaussian (standard results are recalled in the Appendix A). We use for all models the squared exponential kernel with automatic relevance determination (ARD-SE). GP hyperparameters (variance, lengthscales), as well as the noise variance, are

---

[2] Available online as part of the supplementary material of Hill (2011).
[3] https://nda.nih.gov/oai/

estimated with maximum marginal likelihood. Python implementation is carried out with the framework `GPy`[4] (open-source, under BSD licence).

## 5.3 Protocol and evaluation metrics

Our experimental protocol is as follows: each considered dataset is randomly split into a training set $\mathcal{D}$, query set $\mathcal{U}$, and a test set. Experiments mainly focus on the scenario where the test set is a single point $\tilde{\mathbf{x}}$; nonetheless, we also provide additional results for a test population. We then proceed to sequentially acquire $N_{\text{acq}}$ points using the proposed Algorithm 1 and the active learning baselines presented in the next subsection. All experiments are run with $N_{\text{acq}} = 10$.

We track the evolution of two metrics, computed both before the active learning phase and after each acquisition, over $M$ different splits of the original dataset. More precisely, given a split $m$ we track whether the correct decision is returned, with a binary accuracy score

$$A_m = \mathbb{I}(d_{\text{BAYES}}^m, d_\star^m), \tag{14}$$

where $d_{\text{BAYES}}^m$ is the Bayes-optimal decision for $\tilde{\mathbf{x}}_m$ returned by the model (i.e., according to Eq. (2)), and $d_\star^m$ is the ground truth best decision for $\tilde{\mathbf{x}}_m$. We have $\mathbb{I}(d_m, d_m^*) = 1$ if and only if $d_m = d_m^*$ (and zero otherwise). Our second metric is the entropy of the posterior of the optimal decision of the testing point $\tilde{\mathbf{x}}_m$

$$H_m = \mathcal{H}[p(D_{\text{best}}(\tilde{\mathbf{x}}_m)|\mathcal{D}_m)]. \tag{15}$$

All experiments are run with $M = 200$ replications.

## 5.4 Baseline active learning methods

The proposed method (acronym `D-EIG`) is compared with several active learning methods:

- Random sampling (`RS`) – Chooses $(\mathbf{x}_j, d_j)$ uniformly at random from $\mathcal{U}$;

- EIG on the parameters (`P-EIG`) – Presented in Section 2.3. Chooses the $(\mathbf{x}_j, d_j)$ which yields the greatest expected information gain on its associated GP.

- EIG on the outcome (`O-EIG`) – Presented in Section 2.3. Chooses the $(\mathbf{x}_j, d_j)$ which yields the greatest expected information gain on $p(\tilde{y}_{d_j}|\tilde{\mathbf{x}}, \mathcal{D}_{d_j})$. This criterion is connected to the classical expected error reduction criterion, see details in Appendix E.

- Decision uncertainty sampling (`D-US`) – A baseline that we introduce. Chooses the $(\mathbf{x}_j, d_j)$ whose optimal decision (i.e., associated to $\mathbf{x}_j$) is the most uncertain, evaluated with the entropy of $D_{\text{best}}(\mathbf{x}_j)$;

- Uncertainty sampling (`US`) – Chooses the $(\mathbf{x}_j, d_j)$ whose posterior predictive distribution $p(y_{d_j}|\mathbf{x}_j, \mathcal{D}_{d_j})$ has the greatest variance;

## 5.5 Results

Experiments are run with a starting training set of size 100 for the synthetic dataset and the OAI dataset, and of size 50 for the IHDP dataset. All experiments were run on a high-performance computing cluster.

For the scenario with a single test point ($\tilde{\mathbf{x}}$), Figure 3 displays the evolution of the average binary accuracy score $A_m$ over all replications (i.e., the evolution of the proportion of correct decisions). Figure 4 displays the evolution of the average entropy of the posterior of the optimal decision (the $H_m$ score). For all considered datasets, the proposed method gives the best results both in terms of improving the decision-making accuracy and reducing the uncertainty on the optimal decision. This is particularly striking in the OAI dataset, where the problem is the hardest (real data and five possible decisions): all alternatives barely improve the decision-making at all, whereas the proposed method greatly improves it. More precisely, the baselines do

---

[4] https://sheffieldml.github.io/GPy/

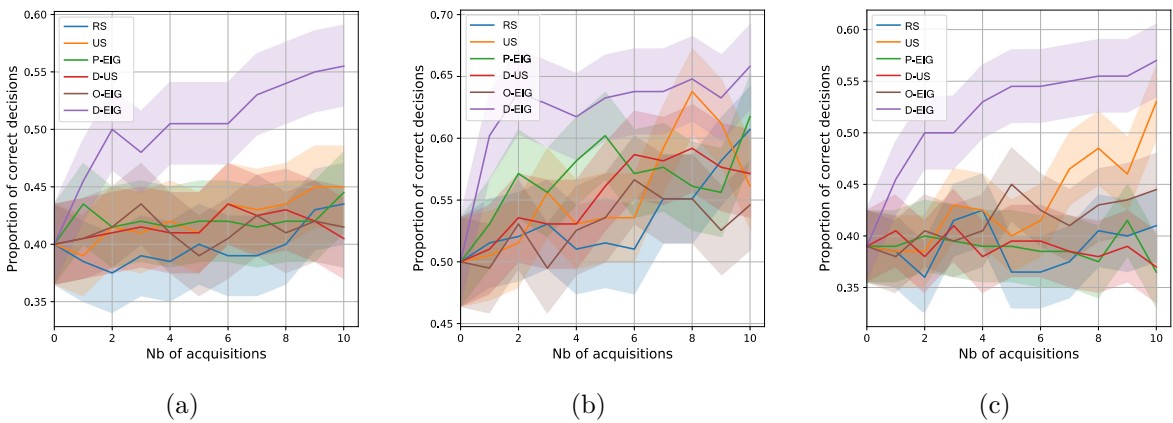

Figure 3: Mean and standard error of the mean of the accuracy score $A_m$ over $M = 200$ replications of the experiment, with a single test point, w.r.t. the number of AL acquisitions. The proposed targeted active learning criterion `D-EIG` outperforms all considered AL methods in improving the accuracy of the decision-making. From left to right: (a) Synthetic data. (b) IHDP dataset. (c) OAI dataset.

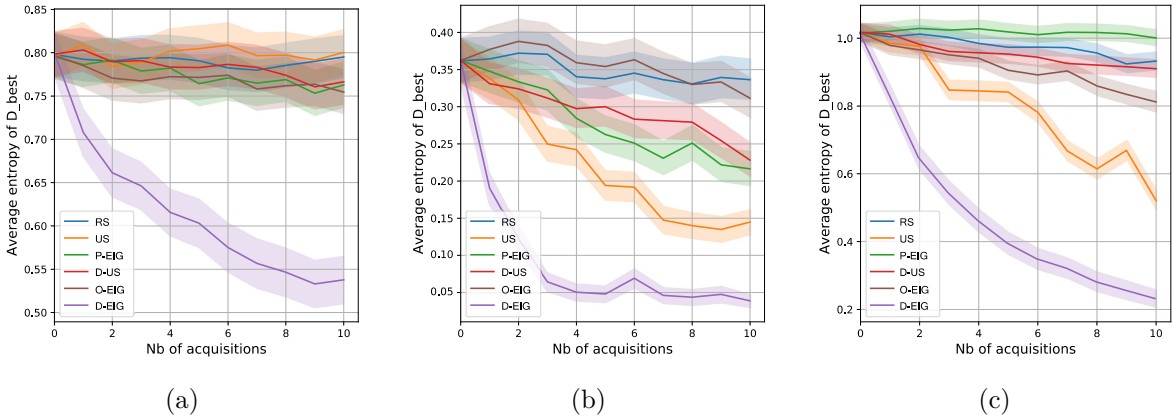

Figure 4: Mean and standard error of the mean of the entropy score $H_m$ (entropy of the posterior of the optimal decision) over $M = 200$ replications of the experiment, with a single test point, w.r.t. the number of AL acquisitions. The proposed targeted active learning criterion `D-EIG` reduces the uncertainty on the optimal decision the fastest among all considered AL methods. From left to right: (a) Synthetic data. (b) IHDP dataset. (c) OAI dataset.

not yield good performance, with the notable exception of `US` which has the second-best performance in entropy reduction on the IHDP and OAI datasets. Lastly, despite being targeted to the outcome $\tilde{\mathbf{x}}$, `O-EIG` has overall poor performance. This demonstrates the value of taking into account the posterior uncertainty on the optimal decision.

For completeness, we also include results with a test population with $N_t = 50$ test points. Figure 5 displays the evolution of the average binary accuracy score $A_m$ over all replications. We draw similar conclusions to the single test point scenario; the proposed method outperforms all other considered baselines.

## 6  Discussion

In this paper, we tackled the problem of decision-making-aware active learning, that is, sample-efficient performance improvement in a down-the-line decision-making problem. To this end, we have proposed to directly reduce the uncertainty on the posterior distribution of the optimal decision. Experimental work

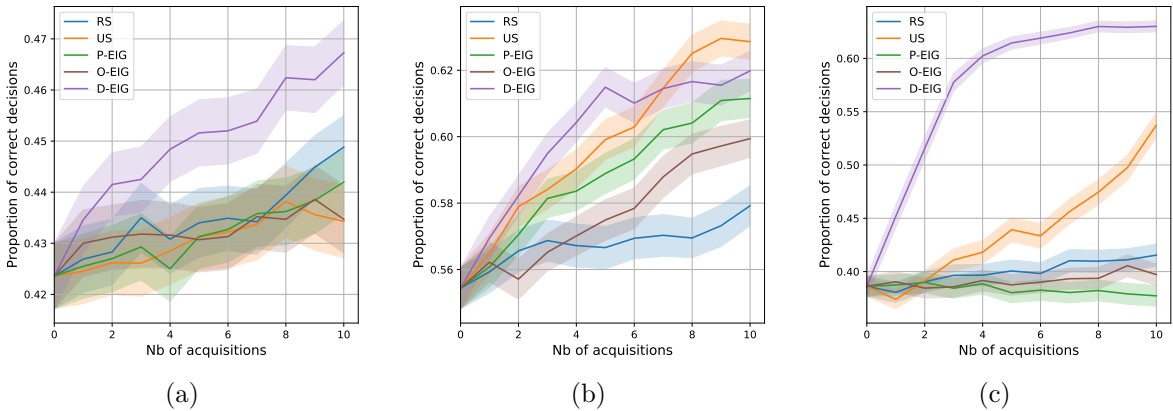

Figure 5: Mean and standard error of the mean of the accuracy score $A_m$ over $M = 200$ replications of the experiment, with a test set of 50 points, w.r.t. the number of AL acquisitions. Here, the proposed targeted active learning criterion `D-EIG` outperforms all considered AL methods in improving the accuracy of the decision-making. From left to right: (a) Synthetic data. (b) IHDP dataset. (c) OAI dataset.

demonstrated the advantages of the proposed technique compared to classical Bayesian experimental design baseline methods in personalized medicine settings.

The main limitation of the proposed method is its computational complexity, as the current implementation involves many model retraining steps. Computational complexity is tolerable in applications where both the utility of correct decisions and cost of acquiring new data points are high, such as in personalized medicine. Nevertheless, future work is needed to design lower-complexity and still accurate approximations of the proposed criterion. A potential direction is to frame the problem as MFBO, as discussed in Section 4, which would enable leveraging of efficient approximation strategies inherent to entropy-search Bayesian optimization (Hennig & Schuler, 2012; Hernández-Lobato et al., 2014; Wang & Jegelka, 2017; Takeno et al., 2020). Extending the proposed criterion to batch selection, in contrast with the current sequential selection method, will also help. The second limitation of our method is that we considered decisions to be available to the algorithm, and in many real-life situations, this may not be the case, for instance, due to privacy concerns. However, our criterion can be straightforwardly extended to tackle this limitation, and we also see this as a direction for future work.

A natural alternative to the proposed active learning criterion is to consider the expected information gain for the maximum utility, instead of focusing on the optimal decision (treatment) that maximizes utility. The rationale behind our proposed criterion is to adhere as closely as possible to the principle: 'Best possible treatment for the patient'. Considering the information gain for the maximizer of the utility (i.e., optimal treatment) implies a criterion that concentrates the active learning budget to identify the optimal treatment, even if the difference in outcomes between two treatments is small. In scenarios where a good treatment is sufficient and it is not necessary to identify the absolutely best one, the criterion that maximizes the expected information gain for maximum utility can be more resource-efficient if two or more treatments work almost equally well for the patient.

To conclude, we anticipate that our method will have a significant impact in the field of interactive AI with healthcare applications. Specifically, we have shown that the proposed technique can be applied in personalized diagnosis and treatment applications. Both of these clinical problems require accurate and reliable decision-making tools, which are, however, costly to build. Our method is sample-efficient and has decision-making capabilities by design.

**Broader Impact Statement**

In the personalized medicine scenario, it would of course be unethical to conduct experiments on other subjects only to gain information for a specific individual. Our perspective in this paper is to retrieve

information from other databases, such as RCTs (meaning that such experiments have already been carried out), or to conduct non-invasive experiments such as asking experts about counterfactuals (Sundin et al., 2019). Nonetheless, building fair active data collection is a crucial direction for research in that field (Andrus et al., 2021). Active learning may conceivably tempt unehtical misuse in personalized medicine, by exposing other patients to threats starting from non-consensual use of information to more sinister scenarios. Special attention is required to prevent misuse while getting the benefits of treatments that active learning promises.

### Acknowledgments

This work was supported by the Academy of Finland (Flagship programme: Finnish Center for Artificial Intelligence, FCAI), grants 292334, 294238, 319264, by the UKRI Turing AI World-Leading Researcher Fellowship, EP/W002973/1, and by the Emil Aaltonen foundation. We also acknowledge the Aalto Science-IT project for their computational resources. LF and AT were affiliated with the Department of Computer Science of Aalto University when this research was conducted. This project was supported by the Research Council of Finland (Profi6 336449 funding program), the University of Oulu strategic funding, and the European Union Horizon Europe - HORIZON-HLTH-2023-STAYHLTH-01 program (grant 101137146).

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

## A    Gaussian process regression

The elements of this section are taken from Rasmussen & Williams (2006), Chapter 2. A Gaussian process (GP) is a stochastic process, i.e., a collection of random variables, such that any finite combination of this collection has a Gaussian distribution. A GP is completely specified by its mean function $m$, and covariance function (or kernel) $\kappa$, and we write

$$f \sim \mathrm{GP}(m(\mathbf{x}), \kappa(\mathbf{x}, \mathbf{x}')). \tag{16}$$

It is often assumed that $m(\mathbf{x}) = 0$.

We now consider a GP regression model

$$f \sim \mathrm{GP}(\mathbf{0}, \kappa(\mathbf{x}, \mathbf{x}')), \tag{17}$$
$$y = f(\mathbf{x}) + \epsilon, \tag{18}$$

where $\epsilon \sim \mathcal{N}(0, \sigma^2)$. That is to say that a GP prior is placed on $f$. In the following, we use the notation $f_{\mathbf{x}}$ to denote $f(\mathbf{x})$. Given a collection of observations $\mathcal{D} = \{(\mathbf{x}_i, y_i)\}_{i=1}^N$, we wish to characterize the posterior distribution at a test point $\tilde{\mathbf{x}}$, $p(f_{\tilde{\mathbf{x}}}|\mathcal{D})$. The definition of a GP implies that

$$\begin{bmatrix} \mathbf{y} \\ f_{\tilde{\mathbf{x}}} \end{bmatrix} \sim \mathcal{N}\left(\mathbf{0}, \begin{bmatrix} \kappa(\mathbf{X}, \mathbf{X}) + \sigma^2\mathbf{I} & \kappa(\mathbf{X}, \tilde{\mathbf{x}}) \\ \kappa(\tilde{\mathbf{x}}, \mathbf{X}) & \kappa(\tilde{\mathbf{x}}, \tilde{\mathbf{x}}) \end{bmatrix}\right), \tag{19}$$

and as such, by using basic manipulations of the Gaussian distribution, it can be shown that

$$p(f_{\tilde{\mathbf{x}}}|\mathcal{D}) = \mathcal{N}(\mu_{\tilde{\mathbf{x}}}, \sigma_{\tilde{\mathbf{x}}}^2), \tag{20}$$

where

$$\mu_{\tilde{\mathbf{x}}} = \kappa(\tilde{\mathbf{x}}, \mathbf{X})[\kappa(\mathbf{X}, \mathbf{X}) + \sigma^2\mathbf{I}]^{-1}\mathbf{y} \tag{21}$$
$$\sigma_{\tilde{\mathbf{x}}}^2 = \kappa(\tilde{\mathbf{x}}, \tilde{\mathbf{x}}) - \kappa(\tilde{\mathbf{x}}, \mathbf{X})[\kappa(\mathbf{X}, \mathbf{X}) + \sigma^2, \mathbf{I}]^{-1}\kappa(\mathbf{X}, \tilde{\mathbf{x}}). \tag{22}$$

Consequently, $p(y|\tilde{\mathbf{x}}, \mathcal{D})$ is also Gaussian with mean $\mu_{\tilde{\mathbf{x}}}$ and variance $\sigma^2 + \sigma_{\tilde{\mathbf{x}}}^2$.

## B    Notes on the expected information gain (EIG)

Let us first consider a standard Bayesian regression model, with likelihood $p(y|\mathbf{x}, \theta)$ and prior $p(\theta)$, which leads to the characterization of the posterior distribution $p(\theta|\mathcal{D})$. We take the example of the EIG on $\theta$. We have

$$\mathrm{EIG}(\mathbf{x}) = \mathcal{H}[p(\theta|\mathcal{D})] - \mathbb{E}_{p(y|\mathbf{x}, \mathcal{D})}\big[\mathcal{H}[p(\theta|\mathcal{D} \cup \{(\mathbf{x}, y)\})]\big]. \tag{23}$$

This expression can be rearranged to show that the EIG is equal to the mutual information between $y$ and $\theta$ (given $\mathbf{x}$ and $\mathcal{D}$), defined as

$$\mathrm{I}(y; \theta|\mathbf{x}, \mathcal{D}) = \iint p(y, \theta|\mathbf{x}, \mathcal{D}) \log \frac{p(y, \theta|\mathbf{x}, \mathcal{D})}{p(y|\mathbf{x}, \mathcal{D})p(\theta|\mathbf{x}, \mathcal{D})} \mathrm{d}y\mathrm{d}\theta. \tag{24}$$

The symmetry of the mutual information leads in turn to an alternative formulation of the EIG, namely

$$\mathrm{EIG}(\mathbf{x}) = \mathcal{H}[p(y|\mathbf{x}, \mathcal{D})] - \mathbb{E}_{p(\theta|\mathcal{D})}[\mathcal{H}[p(y|\mathbf{x}, \theta)]]. \tag{25}$$

which now computes entropies in the output space (and not the parameter space). Most notably, this does not involve model retraining. This is the form most often used in practice.

If we now consider a non-parametric regression model of the form $y = f(\mathbf{x}) + \epsilon$, where $\epsilon \sim \mathcal{N}(0, \sigma^2)$, we can easily adapt the expression of Eq. (25) as

$$\mathrm{EIG}(\mathbf{x}) = \mathcal{H}[p(y|\mathbf{x}, \mathcal{D})] - \mathbb{E}_{p(f|\mathcal{D})}\big[\mathcal{H}[p(y|\mathbf{x}, f)]\big]. \tag{26}$$

This can be further simplified when dealing with GP regression. In that case, the predictive posterior distribution $p(y|\mathbf{x}, \mathcal{D})$ is Gaussian, with mean $\mu_\mathbf{x}$ and variance $\sigma^2 + \sigma_\mathbf{x}^2$. Considering that the value of $\sigma^2$ is fixed, or estimated, the expression of Eq. (26) becomes

$$\text{EIG}(\mathbf{x}) = \frac{1}{2}\left(\log(\sigma_\mathbf{x}^2 + \sigma^2) - \log(\sigma^2)\right). \tag{27}$$

As such, the higher $\sigma_\mathbf{x}^2$, the higher the EIG.

## C   Gauss-Hermite quadrature

We consider computing expectations of the form

$$\mathbb{E}[f(y)] = \int f(y)p(y)\mathrm{d}y, \tag{28}$$

where $Y$ is a Gaussian random variable with mean $\mu$ and variance $\sigma^2$. The Gauss-Hermite approximation of order $N$ of the previous expression is given by

$$\frac{1}{\sqrt{\pi}}\sum_{i=1}^{N}\omega_i f(\sqrt{2}\sigma x_i + \mu), \tag{29}$$

where the $x_i$ are the roots of the Hermite polynomial of order $N$ (denoted by $H_n$), and the weights $\omega_i$ are given by

$$\frac{2^{n-1}n!\sqrt{\pi}}{n^2(H_{n-1}(x_i))^2}. \tag{30}$$

## D   Knee osteoarthritis follow-up data details

We considered all the data from the Osteoarthritis Initiative Dataset (OAI; https://nda.nih.gov/oai/) with a total WOMAC score over 9 (symptomatic subjects). Subsequently, we selected those subjects, which have early, doubtful, or early radiographic Osteoarthritis at the baseline according to the Kellgren Lawrence grading scoring system.

In our experiments, we used a commonly accepted measure – joint space width (JSW) loss over $0.7mm$ as an indicator of progression. The JSW was measured from knee X-rays at a fixed location ($x = 0.250$), thus focusing on OA only in the medial compartment of the knee.

The following is the list of variables, which we selected from the OAI dataset (per knee):

- Age;

- Sex;

- Body-mass-index (BMI);

- Total WOMAC score;

- Indication of varus, valgus, or neither;

- Indication of past injury;

- Indication of past surgery;

- Kellgren-Lawrence grade;

- JSW at a fixed location ($x = 0.250$).

# E  O-EIG and expected error reduction

Let us consider that the error measure is the log-loss. Then, in a Bayesian AL framework, the expected error reduction query writes

$$\mathbf{x}_{\text{EER}} = \arg\min_{\mathbf{x}\in\mathcal{U}} \mathbb{E}_{p(y|\mathbf{x},\mathcal{D})} \left[ \sum_{x_j\in\mathcal{X}_t} \mathcal{H}[p(y_j|x_j, \mathcal{D}\cup\{(\mathbf{x},y)\})] \right], \tag{31}$$

where $\mathcal{X}_t$ is a test population. In the setting considered in the paper, we have $\mathcal{X}_t = \{\tilde{\mathbf{x}}\}$, which reduces to

$$\mathbf{x}_{\text{EER}} = \arg\min_{\mathbf{x}\in\mathcal{U}} \mathbb{E}_{p(y|\mathbf{x},\mathcal{D})} \left[ \mathcal{H}[p(\tilde{y}|\tilde{\mathbf{x}}, \mathcal{D}\cup\{(\mathbf{x},y)\})] \right], \tag{32}$$

which is exactly the O-EIG criterion.

