# OpenReview forum: "Targeted Active Learning for Bayesian Decision-Making"
_TMLR — Accepted by TMLR_

### Review · Reviewer_GuAx · 2024-03-02

**Summary Of Contributions:**

In this paper, the authors have proposed a method for active learning in Bayesian decision-making problems.
A decision-making problem involves selecting an intervention $d$ to maximize the utility $y$ given an input $x$.
The authors considered actively selecting data from unlabeled data $(x, d)$ where the utility $y$ is unknown, to acquire labels for utility $y$.
This approach aims to maximize utility for new inputs $x$ with minimal labeling.
The proposed method adopts uncertainty in decision-making as a criterion for selecting points to acquire labels by estimating the most uncertain data points.
Through experiments on both artificial and real data, the authors have demonstrated that the proposed method outperforms existing active learning methods.
In other words, it effectively improves the accuracy of decision-making with fewer labels.

**Audience:**

Yes

**Broader Impact Concerns:**

There is no ethical concern.

**Claims And Evidence:**

Yes

**Requested Changes:**

See Weakness above.

**Strengths And Weaknesses:**

### Strength: Validity of Proposed Method and Impact of the Results
The proposed method adopts uncertainty in decision-making (9) as a criterion for selecting labeling points.
While conventional active learning methods use parameters or prediction uncertainty as criteria, this criterion is specific to decision-making problems, making it a valid criterion that aligns with the research objectives.
Figures 3 and 4 of the experimental results demonstrate that the proposed method outperforms existing active learning methods in improving decision-making accuracy and uncertainty.
These results indicate that the proposed method is both valid and effective, considering its design for active learning in decision-making.

### Weakness: Uncertainty in Decision-Making and Uncertainty in Maximum Utility
The proposed method adopts uncertainty in decision-making as a criterion for active learning.
However, when two interventions $d$ and $d'$ provide very similar (or equal) utilities $y_d \approx y_{d'}$, determining the superiority of these interventions requires a large amount of data, leading to increased uncertainty.
On the other hand, if the utilities $y_d$ and $y_{d'}$ are nearly equal, the practical impact of choosing either intervention $d$ or $d'$ may be negligible.
In such scenarios, rather than using uncertainty in decision-making as in the proposed method, it might be more appropriate to use uncertainty in the maximum utility $\max_k y_k$ as a criterion, as shown below:

$$
\\mathcal{H}[p(\\max\_k \\tilde{y}\_k | \\tilde{x}, \\mathcal{D} )] - \\mathbb{E}\_{p(y\_j | x\_j, \\mathcal{D}\_{d\_j})} \\mathcal{H}[p(\max\_k \\tilde{y}\_k | \\tilde{x}, \\mathcal{D} \cup \\{(x\_j, d\_j, y\_{d\_j})\\} )]
$$

I presume that one may superior to another one depending on the problem setting.
Adding some discussions on the differences between these two approaches and their respective advantages could further enhance this paper.

---

> ### Author Response · Authors · 2024-04-13
> **Rebuttal to Reviewer GuAx**
>
> We appreciate the reviewer's time and effort reviewing this paper. We discuss the interesting viewpoint raised by the reviewer.
>
> *"The proposed method adopts uncertainty in decision-making as a criterion for active learning. However, when two interventions $d$ and $d'$ provide very similar (or equal) utilities $y_d \approx y_{d'}$, determining the superiority of these interventions requires a large amount of data, leading to increased uncertainty. On the other hand, if the utilities $y_d$ and $y_{d'}$ are nearly equal, the practical impact of choosing either intervention $d$ or $d'$ may be negligible. In such scenarios, rather than using uncertainty in decision-making as in the proposed method, it might be more appropriate to use uncertainty in the maximum utility $\max_k y_k$ as a criterion, as shown below:*
>
> [Equation]
>
> *I presume that one may superior to another one depending on the problem setting. Adding some discussions on the differences between these two approaches and their respective advantages could further enhance this paper."*
>
>  We agree with your interesting observation and the follow-up discussion. We extended our Discussion section by adding the following paragraph: "A natural alternative to the proposed active learning criterion is to consider the expected information gain for the maximum utility, instead of focusing on the optimal decision (treatment) that maximizes utility. The rationale behind our proposed criterion is to adhere as closely as possible to the principle: `Best possible treatment for the patient'. Considering the information gain for the maximizer of the utility (i.e., optimal treatment) implies a criterion that concentrates the active learning budget to identify the optimal treatment, even if the difference in outcomes between two treatments is small. In scenarios where two treatments work almost equally well for the patient, the criterion that maximizes the expected information gain for maximum utility can be more resource-efficient, especially in the long run, as it focuses the active learning budget on identifying the second-best, third-best treatments, etc., rather than being absolutely certain about identifying the optimal one. However, in the early iterations of active learning, our proposed criterion aggressively tries to rule out any potential competitors to the currently believed optimal treatment. Ultimately, in a personalized medicine setting, the priority is finding the best treatment, rather than determining rankings among mediocre treatments.''

---

> > ### Comment · Reviewer_GuAx · 2024-04-15
> > **Re: Rebuttal to Reviewer GuAx**
> >
> > I would like to thank the authors for the detailed reply.
> > I agree that adding a suggested discussion will be helpful for the readers.
> >
> > My question is on the last few lines of the discussion.
> >
> > >  However, in the early iterations of active learning, our proposed criterion aggressively tries to rule out any potential competitors to the currently believed optimal treatment. Ultimately, in a personalized medicine setting, the priority is finding the best treatment, rather than determining rankings among mediocre treatments.
> >
> > If I understand correctly, the criterion based on $\max_k y_k$ will also try to rule out inferior competitors.
> > It will be more convincing if there are some supporting evidences on this point.

---

> > > ### Author Response · Authors · 2024-04-16
> > > **Response to Reviewer GuAx**
> > >
> > > You are right; both criteria will try to rule out competitors, just focusing on different ones (any competitor vs competitors wholes y_k is not close). Thank you for the comment; claims for which method would be faster would indeed need a thorough investigation which would take more time. At this stage we propose to take out that part of the discussion and focus on the part that is clear already from the definitions of the two alternative methods:
> > >
> > > "A natural alternative to the proposed active learning criterion is to consider the expected information gain for the maximum utility, instead of focusing on the optimal decision (treatment) that maximizes utility. The rationale behind our proposed criterion is to adhere as closely as possible to the principle: `Best possible treatment for the patient'. Considering the information gain for the maximizer of the utility (i.e., optimal treatment) implies a criterion that concentrates the active learning budget to identify the optimal treatment, even if the difference in outcomes between two treatments is small. In scenarios where a good treatment is sufficient and it is not necessary to identify the absolutely best one,  the criterion that maximizes the expected information gain for maximum utility can be more resource-efficient if two or more treatments work almost equally well for the patient."

---

> > > > ### Comment · Reviewer_GuAx · 2024-04-18
> > > > **Re: Response to Reviewer GuAx**
> > > >
> > > > I agree that the proposed discussion is helpful and appropriate.

---

### Review · Reviewer_D3ah · 2024-03-08

**Summary Of Contributions:**

This paper considers a variant of the traditional active learning, where the focus is not on generally learning a parameter vector, or non-parametric function as well as possible, but on learning which of $K$ functions $f_1,...f_K$ has the largest expectation at some predefined input vector $\tilde{\mathbf{x}}$. This captures scenarios where a practitioner knows they have a decision to make for a particular individual in the future and seek to efficiently build a model to decide which option (from a discrete set) to choose for the individual.

The main contribution of the paper is a method for this active learning problem, which is a modification of expected information gain based  algorithms, which targets a reduction in the entropy of the (discrete) posterior distribution of the optimal action for $\tilde{\mathbf{x}}$ rather than a posterior on the underlying functions $f_1,...f_K$ or the stochastic output at an arbitrarily selected $\mathbf{x}$. The assessment of the method is empirical, and it compares favourably to a range of sensible benchmarks on a fully synthetic dataset and example datasets from healthcare applications.

There is also brief exploration of the extension of this method to taking decisions for a set of individuals with vectors $\tilde\mathbf{x}_j$, $j=1,...N_t$, where the method can be readily extended (by utilising an upper bound on the entropy in lieu of the computationally intractable exact entropy of the joint distribution) and continues to have an improved performance over benchmarks.

**Audience:**

Yes

**Broader Impact Concerns:**

I believe the current statement is adequate.

**Claims And Evidence:**

Yes

**Requested Changes:**

1. Are we assuming (in our construction of $D_{best}$ and elsewhere) that the best action is always unique? If $f_k: \mathcal{X} \rightarrow \mathcal{S}$ where $\mathcal{S}$ is some discrete set, it would seem that this could quite readily not be the case. Perhaps you intend that the $f's$ are implicitly continuous (as you use a GP to model them) and that the chance of them intersecting at $\tilde{\mathbf{x}}$ is implicitly vanishingly small, but it would be good practice to make this explicit and/or establish a rule for breaking ties.
2. Can you add any detail to convince me that the suggested usage in the healthcare applications is more realistic than it seems?
3. Typos on p5: *cardinality (not cardinal), and the sum in equation (11) should be over $i$ not $t$?
4. On p6, why is the complexity $N_s \times card(\mathcal{U})$ and not $N_s \times card(\mathcal{D})$ or similar? I took it to be the case that we would typically use much fewer than $card(\mathcal{U})$ samples?

It would be ideal to address 1,3, and 4, at least in rebuttal; and 2. is non-essential, but I think worthwhile nonetheless.

**Strengths And Weaknesses:**

Strengths: the problem itself is interesting, and it does seem to be untreated in the literature (surprisingly to me). There is an improved connection to bayesian optimisation (from what I gather there was in a previous iteration), and due reference to related problems in bandits. The empirical evaluation is of a small scale but is sound and shows the method working well. The methodology itself is also very sensible and clearly explained. There are fairly clear directions for future work identified and (some tiny things aside - see below) the work appears to be accurate.

Weaknesses: These are twofold, 1. the size of the contribution (it could have gone further to conduct more extensive evaluation or to provide theoretical guarantees on the method), 2. the connection to applications - I'm not entirely convinced that in the healthcare settings described, this active learning method would actually be used by practitioners and it wouldn't just be more efficient and better for data privacy for a central body to build a predictive model on a larger dataset. A more convincing explanation, a reference to this practice, or an alternative application where the setup feels more commonplace would improve this.

---

> ### Author Response · Authors · 2024-04-13
> **Rebuttal to Reviewer D3ah**
>
> We thank the reviewer for the careful review and appreciate the constructive suggestions. We would appreciate that the reviewer could kindly check our response, which hopefully helps clarify the confusion and resolve the concerns.
>
> *"the connection to applications - I'm not entirely convinced that in the healthcare settings described, this active learning method would actually be used by practitioners and it wouldn't just be more efficient and better for data privacy for a central body to build a predictive model on a larger dataset. A more convincing explanation, a reference to this practice, or an alternative application where the setup feels more commonplace would improve this."*
>
> We note that Reviewer Wi6b has also raised this concern. Please refer to our response to the fourth comment in our rebuttal to Reviewer Wi6b.
>
> *1. Are we assuming (in our construction of $D_{best}$ and elsewhere) that the best action is always unique? If $f_k: \mathcal{X} \rightarrow \mathcal{S}$ where $\mathcal{S}$ is some discrete set, it would seem that this could quite readily not be the case. Perhaps you intend that the $f's$ are implicitly continuous (as you use a GP to model them) and that the chance of them intersecting at $\tilde{\mathbf{x}}$ is implicitly vanishingly small, but it would be good practice to make this explicit and/or establish a rule for breaking ties.*
>
> Indeed, good point. In all the studied cases, the continuity of outcome functions (which are not merely constant functions, etc.) guarantees that the optimal decision is unique almost surely. In theory, the issue of non-uniqueness can also be addressed by assuming that ties are broken non-arbitrarily through a non-stochastic tiebreaker policy. We revised the manuscript by adding a sentence: "We follow the tiebreaker policy that picks the decision with the lowest index. By reindexing the decisions, one can impose a preference for decisions in the rare events of two or more decisions being optimal."
>
> *"2. Can you add any detail to convince me that the suggested usage in the healthcare applications is more realistic than it seems?"*
>
> Please refer to our response to the fourth comment in our rebuttal to Reviewer Wi6b.
>
>  *"3. Typos on p5: \*cardinality (not cardinal), and the sum in equation (11) should be over $i$ not $t$?"*
>
> Thank you for pointing out these typos. We have corrected both for the revised manuscript.
>
> *"4. On p6, why is the complexity $N_s \times card(\mathcal{U})$ and not $N_s \times card(\mathcal{D})$ or similar? I took it to be the case that we would typically use much fewer than $card(\mathcal{U})$ samples?"*
>
> Indeed, this part can be written more clearly. To solve the expected entropy in Equation (10) requires only $N_s$ model re-trainings (one per Monte-Carlo sample). However, in order to solve the optimization problem, i.e. Equation (10), one needs to compute this value for every datapoint $(\mathbf{x},d)$ in the pool $\mathcal{U}$. This leads to the total computational complexity of $N_s \times  \text{card}(\mathcal{U})$. We modified the corresponding part of the text as follows: "All the computational burden resides in the model retraining step, which has to be carried out $N_s \times \text{card}(\mathcal{U})$ times to solve the optimization problem in Eq. (10)."

---

> > ### Comment · Reviewer_D3ah · 2024-04-18
> > **Reply to Authors**
> >
> > Hi authors,
> >
> > Thank you for your responses to my comments, and willingness to modify the areas that could be clearer - I think the clarifications you provide to me will be useful in the paper.
> >
> > Regarding the application in the healthcare setting, it is still somewhat surprising that the employee time constraints prevent the building of a general model, but allow for deployment of personalised model building for individual patients. I am intending to recommend the paper for acceptance, in light of satisfactory replies to other comments and reading the other reviews, but I would recommend making any further additions to justify the authenticity of this example as strongly as possible.
> >
> > Thanks

---

### Review · Reviewer_Wi6b · 2024-04-03

**Summary Of Contributions:**

The authors propose a new active learning approach in the setting where a learner is tasked to make a decision for a single data point that yields the highest return, but before the learner can select a few unlabeled data points and decisions for them and learn about their outcomes (returns). The proposed algorithm is inspired by Bayesian experimental design and aims to take into account the decision-making process and maximize the expected information gain on the posterior distribution of the optimal decision instead of the general quality of predictive performance of the learner. The authors then generalize the framework for the setting of prediction for the population. Finally, the proposed approach is evaluated on 3 different benchmarks, one completely synthetic and two medical datasets against a few baseline active learning methods. The results demonstrate that in both settings (for a single decision and population) and under different budgets for acquiring additional information, the new approach significantly outperforms the baselines.

**Audience:**

Yes

**Broader Impact Concerns:**

I don't have any new concerns in addition to the ones discussed by the authors of the paper - precisely the setting somehow motivates selecting the treatments/making decisions for some patients in such a way as to help gain information that will allow the selection of the best treatment for the other patient. Putting the good of one patient ahead of a few others.

**Claims And Evidence:**

Yes

**Requested Changes:**

None, if more knowledgeable reviewers believe that the current discussion on the related works is now adequate.

**Strengths And Weaknesses:**

Please note that I'm an expert in the topic of Bayesian Active Learning, and as such, it is hard for me to evaluate the novelty of this study and especially the correctness of the discussion of related works properly as I don't know many cited works.


**Strengths:**
- The paper is easy to follow.
- After comparison with the previous submission, it seems that the authors addressed the concerns of the previous reviewers.
- The authors clearly discuss the limitations and specific assumptions of the new setting and the method.
- It seems that the authors consider novel settings for active learning in decision-making, and it is quite well motivated by authors...


**Weaknesses:**
- ... however, it is also very specific. It is hard for me to think about other applications than medicine. As such, I believe this article will be interesting only to a narrow group of readers.
- Under the authors' goal and assumption, the final algorithm is rather simple and computationally very inefficient, but the authors somehow defend this by assuming that the cost of running the algorithm doesn't matter as obtaining additional information for a selected sample is much more expensive than running even very costly algorithm.
- While the authors clearly discuss it, the application of the algorithm may motivate unethical/instrumental treatment of some patients.
- Provided examples of ethical use cases:
  > Our perspective in this paper is to retrieve information from
other databases, such as RCTs (meaning that such experiments have already been carried out), or to conduct non-invasive experiments such as asking experts about counterfactuals (Sundin et al., 2019) (...)

  seems not to meet an assumption about the high cost of obtaining information for new data points. But I belive better examples can be found.

---

> ### Author Response · Authors · 2024-04-13
> **Rebuttal to Reviewer Wi6b**
>
> We are sincerely grateful for the reviewer's appreciation of this paper. For the four comments the reviewer gave,
>
> 1. *“... however, it is also very specific. It is hard for me to think about other applications than medicine. As such, I believe this article will be interesting only to a narrow group of readers.”*
>
> The same problem arises in any targeted decision making task where comprehensive data about relevant other cases is not readily available. Consider investment decisions in business, where various kinds of data are available but identifying relevant ones requires considerable amount of business intelligence work which currently still is predominantly manual and hence costly. Another broad domain is any kind on personalization where cost of privacy loss in accessing others’ data would be appropriately accounted for. Currently people give their privacy to big tech for free but we would very much hope for that to change!
>
> 2. *“Under the authors' goal and assumption, the final algorithm is rather simple and computationally very inefficient, but the authors somehow defend this by assuming that the cost of running the algorithm doesn't matter as obtaining additional information for a selected sample is much more expensive than running even very costly algorithm.”*
>
> Yes indeed, that is out justification for why already the current results are worthwhile. Naturally, further work will be welcome in speeding up the computation. One potential direction is to utilize powerful entropy-search algorithms in BO literature, such as a conceptual MES-PES algorithm discussed in the Related Work section (“Entropy-Search Multi-Fidelity Bayesian Optimization”).
>
> 3. *“While the authors clearly discuss it, the application of the algorithm may motivate unethical/instrumental treatment of some patients.”...“Putting the good of one patient ahead of a few others.”*
>
> We want to emphasize that in the healthcare setting the method is for focusing the resources available for a new patient for that patient. This is not unethical, and while almost any technology can have multiple uses, good and bad, we do think this particular algorithm is worse in that respect than others.
>
> 4. *“Provided examples of ethical use cases:... seems not to meet an assumption about the high cost of obtaining information for new data points. But I belive better examples can be found.”*
>
> We provided a few examples above which hopefully are illustrative as well. Additionally we would like to emphasize that the high cost of obtaining medical information is unfortunately very real. Of the five hospitals we have worked with, only one has well-functioning hospital-wide internal access mechanisms for data, and even those accesses require permissions and hence time from several highly paid employees. Of the two countries we have worked in, one has a very well-accessible national database but it is far from comprehensive in its content and access requires an application process that lasts for weeks and has a moderate financial cost. The other country has a comprehensive database but access is so heavily regulated that in practice it is not available, and even if it was, the administrative costs per patient quickly accumulate to being intolerable for large datasets.
>
> We added a few lines to the current paper to further motivate this discussion (see below). And a more political feature article should actually be written about this dire situation as well.
>
> Added lines / extended discussion in the introduction: “Accessing data from these records involves stringent legal justification due to privacy concerns, mandating that a physician must have a legitimate reason for retrieval and use in patient treatment. The access process can be lengthy and incur significant financial costs, or the administrative costs per patient can quickly accumulate, becoming intolerable for large datasets.”

---

> > ### Comment · Reviewer_Wi6b · 2024-04-23
> >
> > Dear Authors,
> >
> > Thank you for your response, especially for clarifying the cost of obtaining additional information from medical databases. This sounds reasonable, and the proposed addition makes this example more convincing. I suggest adding one more example to strengthen the motivation. But I'm not sure if the one about investment decisions is that convincing, as the method still requires an initial training set, which, in cases like that, may be hard to acquire in general.
> >
> > When it comes to ethical concerns, I believe that an active learning approach in medicine has a greater risk of being improperly used than, let's say, some more standard supervised learning. Especially when we are talking about personalized medicine. I guess we can imagine a dark scenario of some patients being treated in a specific way because it is best for them to help select the best treatment for some wealthy patient. But I, of course, agree that there is nothing unethical about studying such approaches, and I'm happy that there is a discussion on this in the paper.
> >
> > Overall, after reading also the other reviews and discussion, I'm positive about your work.
> >
> > Kind regards

---

> > > ### Author Response · Authors · 2024-05-08
> > > **Response to Reviewer Wi6b**
> > >
> > > We thank you for your response. Below, we address your two comments.
> > >
> > > *"I suggest adding one more example to strengthen the motivation. But I'm not sure if the one about investment decisions is that convincing,..."*
> > > As an example related to an investment decision, consider the problem of loan approval. Suppose a bank wants to make the correct decision on whether to grant a loan to an applicant $x_{test}$. It is possible to access data points $(x,d,y)$ (applicant information, approved/denied, defaulted/repaid) from some records (e.g., the bank’s own or a nationwide database), but each query is costly in terms of time/money or privacy, which implies sensibility of active learning after first potentially collecting an initial training set or having an informative prior on the surrogate model.
> > >
> > > *"When it comes to ethical concerns, I believe that an active learning approach in medicine has a greater risk of being improperly used..."*
> > > We agree, and thought that it is better to make this aspect more explicit in the discussion of ethical concerns. We have extended the discussion in the Broader Impact Statement: "... Active learning may conceivably tempt unehtical misuse in personalized medicine, by exposing other patients to threats starting from non-consensual use of information to more sinister scenarios. Special attention is required to prevent misuse while getting the benefits in treatments that active learning promises."

---

### Decision · Action_Editor_Mz5q · 2024-05-08

**Recommendation:** Accept as is

**Comment:**

All reviewers are recommending acceptance and after reading reviews, rebuttals and the discussion, I certainly concur. Congratulations on a fine piece of work!

**Audience:**

All reviewers agree the paper will be of interest to some of the TMLR audience.

**Claims And Evidence:**

All claims are supported by appropriate evidence.